# RetCL: A Selection-based Approach for Retrosynthesis via Contrastive Learning

## Abstract

Retrosynthesis, of which the goal is to find a set of reactants for synthesizing a target product, is an emerging research area of deep learning. While the existing approaches have shown promising results, they currently lack the ability to consider availability (e.g., stability or purchasability) of the reactants or generalize to unseen reaction templates (i.e., chemical reaction rules). In this paper, we propose a new approach that mitigates the issues by reformulating retrosynthesis into a selection problem of reactants from a candidate set of commercially available molecules. To this end, we design an efficient reactant selection framework, named RetCL (retrosynthesis via contrastive learning), for enumerating all of the candidate molecules based on selection scores computed by graph neural networks. For learning the score functions, we also propose a novel contrastive training scheme with hard negative mining. Extensive experiments demonstrate the benefits of the proposed selection-based approach. For example, when all 671k reactants in the USPTO database are given as candidates, our RetCL achieves top-1 exact match accuracy of $71.3\%$ for the USPTO-50k benchmark, while a recent transformer-based approach achieves $59.6\%$. We also demonstrate that RetCL generalizes well to unseen templates in various settings in contrast to template-based approaches. The code will be released.

## 1 Introduction

*Retrosynthesis* (Corey, 1991), finding a synthetic route starting from commercially available reactants to synthesize a target product (see Figure 1a), is at the center of focus for discovering new materials in both academia and industry. It plays an essential role in practical applications by finding a new synthetic path, which can be more cost-effective or avoid patent infringement. However, retrosynthesis is a challenging task that requires searching over a vast number of molecules and chemical reactions, which is intractable to enumerate. Nevertheless, due to its utter importance, researchers have developed computer-aided frameworks to automate the process of retrosynthesis for more than three decades (Corey et al., 1985).

The computer-aided approaches for retrosynthesis mainly fall into two categories depending on their reliance on the reaction templates, i.e., sub-graph patterns describing how the chemical reaction occurs among reactants (see Figure 1b). The template-based approaches (Coley et al., 2017b; Segler & Waller, 2017; Dai et al., 2019) first enumerate known reaction templates and then apply a well-matched template into the target product to obtain reactants. Although they can provide chemically interpretable predictions, they limit the search space to known templates and cannot discover novel synthetic routes. In contrast, template-free approaches (Liu et al., 2017; Karpov et al., 2019; Zheng et al., 2019; Shi et al., 2020) generate the reactants from scratch to avoid relying on the reaction templates. However, they require to search the entire molecular space, and their predictions could be either unstable or commercially unavailable.

We emphasize that retrosynthesis methods are often required to consider the availability of reactants and generalize to unseen templates in real-world scenarios. For example, when a predicted reactant is not available (e.g., not purchasable) for a chemist or a laboratory, the synthetic path starting from the predicted reactant cannot be instantly used in practice. Moreover, chemists often require retrosynthetic analysis based on unknown reaction rules. This is especially significant due to our

(a) Chemical reaction        (b) Reaction template

Figure 1: Examples of (a) a chemical reaction and (b) the corresponding reaction template in the USPTO-50k dataset. The objective of retrosynthesis is to find the reactants for the given product.

incomplete knowledge of chemical reactions; e.g., 29 million reactions were regularly recorded between 2009 and 2019 in Reaxys[1] (Mutton & Ridley, 2019).

**Contribution.** In this paper, we propose a new *selection-based* approach, which allows considering the commercial availability of reactants. To this end, we reformulate the task of retrosynthesis as a problem where reactants are selected from a candidate set of available molecules. This approach has two benefits over the existing ones: (a) it guarantees the commercial availability of the selected reactants, which allows chemists proceeding to practical procedures such as lab-scale experiments or optimization of reaction conditions; (b) it can generalize to unseen reaction templates and find novel synthetic routes.

For the selection-based retrosynthesis, we propose an efficient selection framework, named RETCL (retrosynthesis via contrastive learning). To this end, we design two effective selection scores in synthetic and retrosynthetic manners. To be specific, we use the cosine similarity between molecular embeddings of the product and the reactants computed by graph neural networks. For training the score functions, we also propose a novel contrastive learning scheme (Sohn, 2016; He et al., 2019; Chen et al., 2020b) with hard negative mining (Harwood et al., 2017) to overcome a scalability issue while handling a large-scale candidate set.

To demonstrate the effectiveness of our RETCL, we conduct various experiments based on the USPTO database (Lowe, 2012) containing 1.8M chemical reactions in the US patent literature. Thanks to our prior knowledge on the candidate reactants, our method achieves $71.3\%$ test accuracy and significantly outperforms the baselines without such prior knowledge. Furthermore, our algorithm demonstrates its superiority even when enhancing the baselines with candidate reactants, e.g., our algorithm improves upon the existing template-free approach (Chen et al., 2019) by $11.7\%$. We also evaluate the generalization ability of RETCL by testing USPTO-50k-trained models on the USPTO-full dataset; we obtain $39.9\%$ test accuracy while the state-of-the-art template-based approach (Dai et al., 2019) achieves $26.7\%$. Finally, we demonstrate how our RETCL can improve multi-step retrosynthetic analysis where intermediate reactants are not in our candidate set.

We believe our scheme has the potential to improve further in the future, by utilizing (a) additional chemical knowledge such as atom-mapping or leaving groups (Shi et al., 2020; Somnath et al., 2020); (b) various contrastive learning techniques in other domains, e.g., computer vision (He et al., 2019; Chen et al., 2020b; Hénaff et al., 2019; Tian et al., 2019), audio processing (Oord et al., 2018), and reinforcement learning (Srinivas et al., 2020).

## 2   SELECTION-BASED RETROSYNTHESIS VIA CONTRASTIVE LEARNING

### 2.1   OVERVIEW OF RETCL

In this section, we propose a selection framework for retrosynthesis via contrastive learning, coined RETCL. Our framework is based on solving the retrosynthesis task as a selection problem over a candidate set of *commercially available reactants* given the target product. Especially, we design a selection procedure based on molecular embeddings computed by graph neural networks and train the networks via contrastive learning.

To this end, we define a chemical reaction $\mathcal{R} \rightarrow P$ as a synthetic process of converting a *reactant-set* $\mathcal{R} = \{R_1, \ldots, R_n\}$, i.e., a set of *reactant* molecules, to a *product* molecule $P$ (see Figure 1a). We

---

[1]A chemical database, https://www.reaxys.com

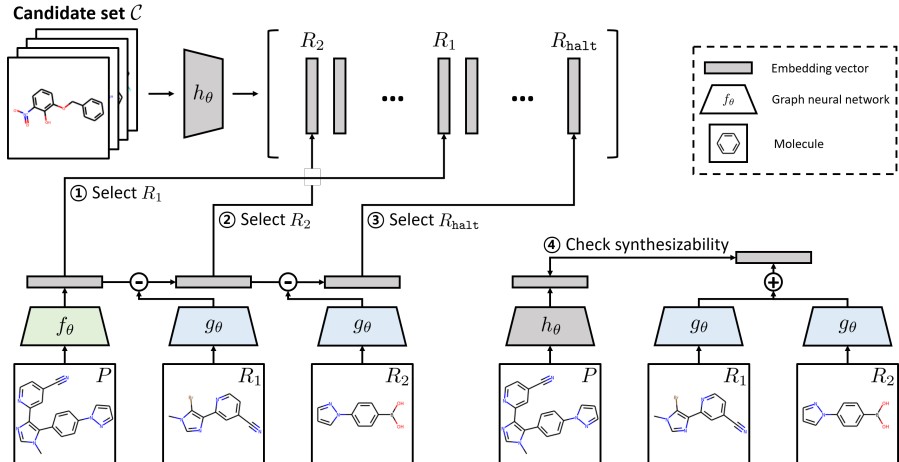

Figure 2: Illustration of the search procedure in RETCL. It first (1-3) selects reactants sequentially based on $\psi(R|P, \mathcal{R}_{\texttt{given}})$, and then (4) check the synthesizability of the selected reactant-set based on $\phi(P|\mathcal{R})$. The overall score is the average over all scores from (1) to (4).

aim to solve the problem of retrosynthesis by finding the reactant-set $\mathcal{R}$ from a *candidate set* $\mathcal{C}$ which can be synthesized to the target product $P$. Especially, we consider the case when the candidate set $\mathcal{C}$ consists of *commercially available* molecules. Throughout this paper, we say that the synthetic direction (from $\mathcal{R}$ to $P$) is *forward* and the retrosynthetic direction (from $P$ to $\mathcal{R}$) is *backward*.

Note that our framework stands out from the existing works in terms of the candidate set $\mathcal{C}$. To be specific, (a) template-free approaches (Lin et al., 2019; Karpov et al., 2019; Shi et al., 2020) choose $\mathcal{C}$ as the whole space of (possibly unavailable) molecules; and (b) template-based approaches (Coley et al., 2017b; Segler & Waller, 2017; Dai et al., 2019) choose $\mathcal{C}$ as possible reactants extracted from the known reaction templates. In comparison, our framework neither requires (a) search over the entire space of molecules, or (b) domain knowledge to extract the reaction templates.

We now briefly outline the RETCL framework. Our framework first searches the most likely reactant-sets $\mathcal{R}_1, \ldots, \mathcal{R}_T \subset \mathcal{C}$ in a sequential manner based on a backward selection score $\psi(R|P, \mathcal{R}_{\texttt{given}})$, and then ranks the reactant-sets using $\psi(R|P, \mathcal{R}_{\texttt{given}})$ and another forward score $\phi(P|\mathcal{R})$. For learning the score functions, we propose a novel contrastive learning scheme with hard negative mining for improving the selection qualities. We next provide detailed descriptions of the search procedure and the training scheme in Section 2.2 and 2.3, respectively.

## 2.2 SEARCH PROCEDURE WITH GRAPH NEURAL NETWORKS

We first introduce the search procedure of RETCL in detail. To find a reactant-set $\mathcal{R} = \{R_1, \ldots, R_n\}$, we select each element $R_i$ sequentially from the candidate set $\mathcal{C}$ based on the *backward* (retrosynthetic) selection score $\psi(R|P, \mathcal{R}_{\texttt{given}})$. It represents a selection score of a reactant $R$ given a target product $P$ and a set of previously selected reactants $\mathcal{R}_{\texttt{given}} \subset \mathcal{C}$. Note that the score function is also capable of selecting a special reactant $R_{\texttt{halt}}$ to stop updating the reactant-set. Using beam search, we choose top $T$ likely reactant-sets $\mathcal{R}_1, \ldots, \mathcal{R}_T$.

Furthermore, we rank the chosen reactant-sets $\mathcal{R}_1, \ldots, \mathcal{R}_T$ based on the backward selection score $\psi(R|P, \mathcal{R}_{\texttt{given}})$ and the *forward* (synthetic) score $\phi(P|\mathcal{R})$. The latter represents the synthesizability of $\mathcal{R}$ for $P$. Note that $\psi(R|P, \mathcal{R}_{\texttt{given}})$ and $\phi(P|\mathcal{R})$ correspond to backward and forward directions of a chemical reaction $\mathcal{R} \rightarrow P$, respectively (see Section 2.1 and Figure 1a). Using both score functions, we define an overall score on a chemical reaction $\mathcal{R} \rightarrow P$ as follows:

$$\texttt{score}(P, \mathcal{R}) = \frac{1}{n+2}\left(\max_{\pi \in \Pi} \sum_{i=1}^{n+1} \psi(R_{\pi(i)}|P, \{R_{\pi(1)}, \ldots, R_{\pi(i-1)}\}) + \phi(P|\mathcal{R})\right), \quad (1)$$

where $R_{n+1} = R_{\texttt{halt}}$ and $\Pi$ is the space of permutations defined on the integers $1, \ldots, n+1$ satisfying $\pi(n+1) = n+1$. Based on $\texttt{score}(P, \mathcal{R})$, we decide the rankings of $\mathcal{R}_1, \ldots, \mathcal{R}_T$ for synthesizing

the target product $P$. We note that the $\max_{\pi \in \Pi}$ operator and the $\frac{1}{n+2}$ term make the overall score (equation 1) be independent of order and number of reactants, respectively. Figure 2 illustrates this search procedure of our framework.

**Score design.** We next elaborate our design choices for the score functions $\psi$ and $\phi$. We first observe that the molecular graph of the product $P$ can be decomposed into subgraphs from each reactant of the reactant-set $\mathcal{R}$, as illustrated in Figure 1a. Moreover, when selecting reactants sequentially, the structural information of the previously selected reactants $\mathcal{R}_{\texttt{given}}$ should be ignored to avoid duplicated selections. From these observations, we design the scores $\psi_\theta(R|P, \mathcal{R}_{\texttt{given}})$ and $\phi(P|\mathcal{R})$ as follows:

$$\psi(R|P, \mathcal{R}_{\texttt{given}}) = \texttt{CosSim}\left(f_\theta(P) - \sum\nolimits_{S \in \mathcal{R}_{\texttt{given}}} g_\theta(S),\ h_\theta(R)\right),$$

$$\phi(P|\mathcal{R}) = \texttt{CosSim}\left(\sum\nolimits_{R \in \mathcal{R}} g_\theta(R),\ h_\theta(P)\right),$$

where $\texttt{CosSim}$ is the cosine similarity and $f_\theta, g_\theta, h_\theta$ are embedding functions from a molecule to a fixed-sized vector with parameters $\theta$. Note that one could think that $f_\theta$ and $g_\theta$ are query functions for a product and a reactant, respectively, while $h_\theta$ is a key function for a molecule. Such a query-key separation allows the search procedure to be processed as an efficient matrix-vector multiplication. This computational efficiency is important in our selection-based setting because the number of candidates is often very large, e.g., $|\mathcal{C}| \approx 6 \times 10^5$ for the USPTO dataset.

To parameterize the embedding functions $f_\theta$, $g_\theta$ and $h_\theta$, we use the recently proposed graph neural network (GNN) architecture, structure2vec (Dai et al., 2016; 2019). The implementation details of the architecture is described in Section 3.1.

**Incorporating reaction types.** A human expert could have some prior information about a reaction type, e.g., carbon-carbon bond formation, for the target product $P$. To utilize this prior knowledge, we add trainable embedding bias vectors $u^{(t)}$ and $v^{(t)}$ for each reaction type $t$ into the query embeddings of $\psi$ and $\phi$, respectively. For example, $\phi(P|\mathcal{R})$ becomes $\texttt{CosSim}(\sum_{R \in \mathcal{R}} g_\theta(R) + v^{(t)},\ h_\theta(P))$. The bias vectors are initialized by zero at beginning of training.

## 2.3 TRAINING SCHEME WITH CONTRASTIVE LEARNING

Finally, we describe our learning scheme for training the score functions defined in Section 2.1 and 2.2. We are inspired by how the score functions $\psi(R|P, \mathcal{R}_{\texttt{given}})$ and $\phi(P|\mathcal{R})$ resemble the classification scores of selecting (a) the reactant $R$ given the product $P$ and the previously selected reactants $\mathcal{R}_{\texttt{given}}$ and (b) the product $P$ given all of the selected reactants $\mathcal{R}$, respectively. Based on this intuition, we consider two classification tasks with the following probabilities:

$$p(R|P, \mathcal{R}_{\texttt{given}}, \mathcal{C}) = \frac{\exp(\psi(R|P, \mathcal{R}_{\texttt{given}})/\tau)}{\sum_{R' \in \mathcal{C} \setminus \{P\}} \exp(\psi(R'|P, \mathcal{R}_{\texttt{given}})/\tau)},$$

$$q(P|\mathcal{R}, \mathcal{C}) = \frac{\exp(\phi(P|\mathcal{R})/\tau)}{\sum_{P' \in \mathcal{C} \setminus \mathcal{R}} \exp(\phi(P'|\mathcal{R})/\tau)},$$

where $\tau$ is a hyperparameter for temperature scaling and $\mathcal{C}$ is the given candidate set of molecules. Note that we do not consider $P$ and $R \in \mathcal{R}$ as available reactants and products for the classification tasks of $p$ and $q$, respectively. This reflects our prior knowledge that the product $P$ is always different from the reactants $\mathcal{R}$ in a chemical reaction. As a result, we arrive at the following losses defined on a reaction of the product $P$ and the reactant-set $\mathcal{R} = \{R_1, \ldots, R_n\}$:

$$\mathcal{L}_{\texttt{backward}}(P, \mathcal{R}|\theta, \mathcal{C}) = -\max_{\pi \in \Pi} \sum_{i=1}^{n+1} \log p(R_{\pi(i)}|P, \{R_{\pi(1)}, \ldots, R_{\pi(i-1)}\}, \mathcal{C}),$$

$$\mathcal{L}_{\texttt{forward}}(P, \mathcal{R}|\theta, \mathcal{C}) = -\log q(P|\mathcal{R}, \mathcal{C}),$$

where $R_{n+1} = R_{\texttt{halt}}$ and $\Pi$ is the space of permutations defined on the integers $1, \ldots, n+1$ satisfying $\pi(n+1) = n+1$. We note that minimizing the above losses increases the scores $\psi(R|P, \mathcal{R}_{\texttt{given}})$ and $\phi(P|\mathcal{R})$ of the correct pairs of product and reactants, i.e., numerators, while decreasing that of wrong pairs, i.e., denominators. Such an objective is known as contrastive loss which has recently gained

much attention in various domains (Sohn, 2016; He et al., 2019; Chen et al., 2020b; Oord et al., 2018; Srinivas et al., 2020).

Unfortunately, the optimization of $\mathcal{L}_{\texttt{backward}}$ and $\mathcal{L}_{\texttt{forward}}$ is intractable since the denominators of $p(R|P, \mathcal{R}_{\texttt{given}}, \mathcal{C})$ and $q(P|\mathcal{R}, \mathcal{C})$ require summation over the large set of candidate molecules $\mathcal{C}$. To resolve this, for each mini-batch of reactions $\mathcal{B}$ sampled from the training dataset, we approximate $\mathcal{C}$ with the following set of molecules:

$$\mathcal{C}_{\mathcal{B}} = \{M \mid \exists\, (\mathcal{R}, P) \in \mathcal{B} \text{ such that } M = P \text{ or } M \in \mathcal{R}\},$$

i.e., $\mathcal{C}_{\mathcal{B}}$ is the set of all molecules in $\mathcal{B}$. Then we arrive at the following training objective:

$$\mathcal{L}(\mathcal{B}|\theta) = \frac{1}{|\mathcal{B}|} \sum_{(\mathcal{R},P)\in\mathcal{B}} \left( \mathcal{L}_{\texttt{backward}}(P, \mathcal{R}|\theta, \mathcal{C}_{\mathcal{B}}) + \mathcal{L}_{\texttt{forward}}(P, \mathcal{R}|\theta, \mathcal{C}_{\mathcal{B}}) \right). \quad (2)$$

**Hard negative mining.** In our setting, molecules in the candidate set $\mathcal{C}_{\mathcal{B}}$ are easily distinguishable. Hence, learning to discriminate between them is often not informative. To alleviate this issue, we replace the $\mathcal{C}_{\mathcal{B}}$ with its augmented version $\widetilde{\mathcal{C}}_{\mathcal{B}}$ by adding *hard* negative samples, i.e., similar molecules, as follows:

$$\widetilde{\mathcal{C}}_{\mathcal{B}} = \mathcal{C}_{\mathcal{B}} \cup \bigcup_{M\in\mathcal{C}_{\mathcal{B}}} \{\text{Top-}K \text{ nearest neighbors of } M \text{ from } \mathcal{C}\},$$

where $K$ is a hyperparameter controlling hardness of the contrastive task. The nearest neighbors are defined with respect to the cosine similarity on $\{h_\theta(M)\}_{M\in\mathcal{C}}$. Since computing all embeddings $\{h_\theta(M)\}_{M\in\mathcal{C}}$ for every iteration is time-consuming, we update information of the nearest neighbors periodically. We found that the hard negative mining plays a significant role in improving the performance of RETCL (see Section 3.3).

## 3 EXPERIMENTS

### 3.1 EXPERIMENTAL SETUP

**Dataset.** We mainly evaluate our framework in USPTO-50k, which is a standard benchmark for the task of retrosynthesis. It contains 50k reactions of 10 reaction types derived from the US patent literature, and we divide it into training/validation/test splits following Coley et al. (2017b). To apply our framework, we choose the candidate set of commercially available molecules $\mathcal{C}$ as the all reactants in the entire USPTO database as Guo et al. (2020) did. This results in the candidate set with a size of 671,518. For the evaluation metric, we use the top-$k$ exact match accuracy, which is widely used in the retrosynthesis literature. We also experiment with other USPTO benchmarks for more challenging tasks, e.g., generalization to unseen templates. We provide a more detailed description of the USPTO benchmarks in Appendix A.

**Hyperparameters.** We use a single shared 5-layer structure2vec (Dai et al., 2016; 2019) architecture and three separate 2-layer residual blocks with an embedding size of 256. To obtain graph-level embedding vectors, we use `sum` pooling over `mean` pooling since it captures the size information of molecules. For contrastive learning, we use a temperature of $\tau = 0.1$ and $K = 4$ nearest neighbors for hard negative mining. More details are provided in Appendix B.

### 3.2 SINGLE-STEP RETROSYNTHESIS IN USPTO-50K

Table 1 evaluates our RETCL and other baselines using the top-$k$ exact match accuracy with $k \in \{1, 3, 5, 10, 20, 50\}$. We first note that our framework significantly outperforms a concurrent selection-based approach,[2] Bayesian-Retro (Guo et al., 2020), by $23.8\%$ and $23.7\%$ in terms of top-1 accuracy when reaction type is unknown and given, respectively. Furthermore, ours also outperforms template-based approaches utilizing the different knowledge, i.e., reaction templates instead of candidates, with a large margin, e.g., $18.8\%$ over GLN (Dai et al., 2019) in terms of top-1 accuracy when reaction type is unknown.

---

[2] Note that Bayesian-Retro (Guo et al., 2020) is not scalable to a large candidate set. See Section 4 for details.

Table 1: The top-$k$ exact match accuracy (%) of computer-aided approaches in USPTO-50k. The template-based approaches use the knowledge of reaction templates while others do not. [†]The results are reproduced using the code of Chen et al. (2019).

| Category | Method | Top-1 | Top-3 | Top-5 | Top-10 | Top-20 | Top-50 |
|---|---|---|---|---|---|---|---|
| **Reaction type is unknown** | | | | | | | |
| Template-free | Transformer (Karpov et al., 2019) | 37.9 | 57.3 | 62.7 | - | - | - |
| | SCROP (Zheng et al., 2019) | 43.7 | 60.0 | 65.2 | 68.7 | - | - |
| | Transformer (Chen et al., 2019) | 44.8 | 62.6 | 67.7 | 71.1 | - | - |
| | G2Gs (Shi et al., 2020) | **48.9** | **67.6** | **72.5** | **75.5** | - | - |
| Template-based | retrosim (Coley et al., 2017b) | 37.3 | 54.7 | 63.3 | 74.1 | 82.0 | 85.3 |
| | neuralsym (Segler & Waller, 2017) | 44.4 | 65.3 | 72.4 | 78.9 | 82.2 | 83.1 |
| | GLN (Dai et al., 2019) | **52.5** | **69.0** | **75.6** | **83.7** | **89.0** | **92.4** |
| Selection-based | Bayesian-Retro (Guo et al., 2020) | 47.5 | 67.2 | 77.0 | 80.3 | - | - |
| | RETCL (Ours) | **71.3** | **86.4** | **92.0** | **94.1** | **95.0** | **96.4** |
| **Reaction type is given as prior** | | | | | | | |
| Template-free | seq2seq (Liu et al., 2017) | 37.4 | 52.4 | 57.0 | 61.7 | 65.9 | 70.7 |
| | Transformer[†] (Chen et al., 2019) | 54.1 | 70.0 | 74.2 | 77.8 | 80.4 | 83.3 |
| | SCROP (Zheng et al., 2019) | 59.0 | 74.8 | 78.1 | 81.1 | - | - |
| | G2Gs (Shi et al., 2020) | **61.0** | **81.3** | **86.0** | **88.7** | - | - |
| Template-based | retrosim (Coley et al., 2017b) | 52.9 | 73.8 | 81.2 | 88.1 | 91.8 | 92.9 |
| | neuralsym (Segler & Waller, 2017) | 55.3 | 76.0 | 81.4 | 85.1 | 86.5 | 86.9 |
| | GLN (Dai et al., 2019) | **64.2** | **79.1** | **85.2** | **90.0** | **92.3** | **93.2** |
| Selection-based | Bayesian-Retro (Guo et al., 2020) | 55.2 | 74.1 | 81.4 | 83.5 | - | - |
| | RETCL (Ours) | **78.9** | **90.4** | **93.9** | **95.2** | **95.8** | **96.7** |

Table 2: The top-$k$ exact match accuracy (%) of our RETCL, Transformer (Chen et al., 2019) and GLN (Dai et al., 2019) with discarding predictions not in the candidate set $\mathcal{C}$.

| Category | Method | Top-1 | Top-5 | Top-10 | Top-50 | Top-100 | Top-200 |
|---|---|---|---|---|---|---|---|
| **Reaction type is unknown** | | | | | | | |
| Template-free | Transformer (Chen et al., 2019) | 59.6 | 74.3 | 77.0 | 79.4 | 79.5 | 79.6 |
| | RETCL (Ours) | 71.3 | **92.0** | **94.1** | **96.4** | **96.7** | **97.1** |
| Template-based | GLN (Dai et al., 2019) | **77.3** | 90.0 | 92.5 | 93.3 | 93.3 | 93.3 |
| **Reaction type is given as prior** | | | | | | | |
| Template-free | Transformer (Chen et al., 2019) | 68.4 | 82.4 | 84.3 | 85.9 | 86.0 | 86.1 |
| | RETCL (Ours) | 78.9 | **93.9** | **95.2** | **96.7** | **97.1** | **97.5** |
| Template-based | GLN (Dai et al., 2019) | **82.0** | 91.7 | 92.9 | 93.3 | 93.3 | 93.3 |

**Incorporating the knowledge of candidates into baselines.** However, it is hard to fairly compare between methods operating under different assumptions. For example, template-based approaches require the knowledge of reaction templates, while our selection-based approach requires that of available reactants. To alleviate such a concern, we incorporate our prior knowledge of candidates $\mathcal{C}$ into the baselines; we filter out reactants outside the candidates $\mathcal{C}$ from the predictions made by the baselines. As reported in Table 2, our framework still outperforms the template-free approaches with a large margin, e.g., Transformer (Chen et al., 2019) achieves 68.4% in the top-1 accuracy, while we achieve 78.9% when reaction type is given. Although GLN uses more knowledge than ours in this setting, its top-$k$ accuracy is saturated to 93.3% which is the coverage of known templates, i.e., the upper bound of template-based approaches. However, our framework continues to increase the top-$k$ accuracy as $k$ increases, e.g., 97.5% in terms of top-200 accuracy. We additionally compare with SCROP (Zheng et al., 2019) using their publicly available predictions with reaction types; SCROP achieves 70.4% in the top-1 accuracy, which also underperforms ours.

## 3.3 ANALYSIS AND ABLATION STUDY

**Failure cases.** Figure 3 shows examples of wrong predictions from our framework. We found that the reactants of wrong predictions are still similar to the ground-truth ones. For example, the top-3 predictions of the examples A and B are partially correct; the larger reactant is correct while the

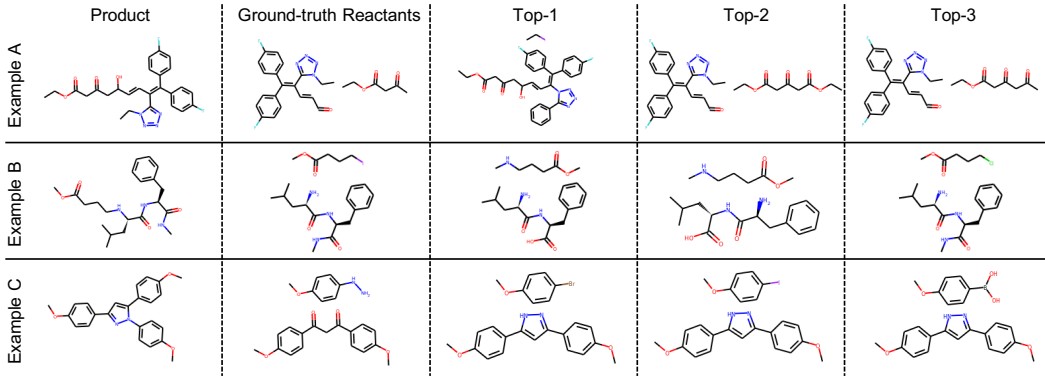

Figure 3: Failure cases of RETCL.

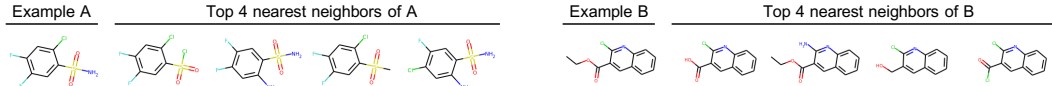

Figure 4: The top-4 nearest neighbors of two randomly sampled molecules in $\mathcal{C}$.

smaller one is slightly different. In the example C, the ring at the center of the product is broken in the ground-truth reactants while our RETCL predicts non-broken reactants. Surprisingly, in a chemical database, Reaxys, we found a synthetic route starting from reactants in the top-2 prediction to synthesize the target product. We attach the corresponding route to Appendix C. These results show that our RETCL could provide meaningful information for retrosynthetic analysis in practice.

**Nearest neighbors on molecular embeddings.** For hard negative mining described in Section 2.3, it is required to find similar molecules using the cosine similarity on $\{h_\theta(M)\}_{M \in \mathcal{C}}$. As illustrated in Figure 4, $h_\theta(M)$ is capable of capturing the molecular structures.

**Effect of components.** Table 3 shows the effect of components of our framework. First, we found that the hard negative mining as described in Section 2.3 increases the performance significantly. This is because there are many similar molecules in the candidate set $\mathcal{C}$, thus a model could predict slightly different reactants without hard negative mining. We also demonstrate the effect of checking the synthesizablity of the predicted reactants with $\phi(P|\mathcal{R})$. As seen the fourth and fifth rows in Table 3, using

Table 3: Ablation study.

| $\phi(P|\mathcal{R})$ | $K$ | sum | Top-1 | Top-10 |
|---|---|---|---|---|
| ✓ | | | 59.5 | 79.8 |
| ✓ | 1 | | 69.6 | 92.2 |
| ✓ | 2 | | 70.9 | 92.7 |
| ✓ | 4 | | 71.1 | 92.9 |
| | 4 | | 69.8 | 90.3 |
| ✓ | 4 | ✓ | 71.3 | 94.1 |

$\phi(P|\mathcal{R})$ provides a 2.6% gain in terms of top-10 accuracy. Moreover, we empirically found that sum pooling for aggregating node embedding vectors is more effective than mean pooling. This is because the former can capture the size of molecules as the norm of graph embedding vectors.

## 3.4 MORE CHALLENGING RETROSYNTHESIS TASKS

**Generalization to unseen templates.** The advantage of our framework over the template-based approaches is the generalization ability to unseen reaction templates. To demonstrate it, we remove reactions of classes (i.e., reaction types) from 6 to 10 in training/validation splits of the USPTO-50k benchmark. Then the number of remaining reactions is 27k. In this case, the templates extracted from the modified dataset cannot be applied to the reactions of different classes. Hence the template-based approaches suffer from the generalization issue; for example, GLN (Dai et al., 2019) cannot provide correct predictions for reactions of unseen types as reported in Table 4, while our RETCL is able to provide correct answers.

We also conduct a more realistic experiment: testing on a larger dataset, the test split of USPTO-full dataset preprocessed by Dai et al. (2019), using a model trained on a smaller dataset, USPTO-50k. We note that the number of reactions for training, 40k, is smaller than that of testing reac-

Table 5: Generalization to USPTO-full.

| Method | Top-1 | Top-10 | Top-50 |
|---|---|---|---|
| Transformer (Chen et al., 2019) | 29.9 | 46.6 | 51.0 |
| GLN (Dai et al., 2019) | 26.7 | 42.2 | 46.7 |
| RETCL (Ours) | 39.9 | 57.1 | 60.9 |

Table 4: The top-10 exact match accuracy (%) of our RETCL and GLN (Dai et al., 2019) trained on USPTO-50k without reaction types from 6 to 10. The average column indicates the average of class-wise accuracy for each reaction type.

| Method | Average | Reaction type | | | | | | | | | |
|---|---|---|---|---|---|---|---|---|---|---|---|
| | | 1 | 2 | 3 | 4 | 5 | 6 | 7 | 8 | 9 | 10 |
| GLN (Dai et al., 2019) | 39.7 | 84.3 | 92.2 | 70.7 | 59.3 | 89.7 | 0.0 | 0.0 | 0.0 | 0.5 | 0.0 |
| RETCL (Ours) | **55.6** | **93.9** | **97.6** | **86.4** | **67.0** | **95.6** | **59.1** | **11.9** | **18.3** | **26.1** | 0.0 |

tions, 100k. As reported in Table 5, our framework provides a consistent benefit over the template-based approaches. These results show that our strength of generalization ability.

**Multi-step retrosynthesis.** To consider a more practical scenario, we evaluate our algorithm for the task of *multi-step retrosynthesis*. To this end, we use the synthetic route benchmark provided by Chen et al. (2020a). Here, we assume that only the building blocks (or starting materials)

Table 6: Multi-step retrosynthesis.

| Single-step model | Single | | Hybrid | |
|---|---|---|---|---|
| | MLP | TF | TF+TF | RETCL+TF |
| Succ. rate (%) | 86.84 | 91.05 | 90.54 | 96.84 |
| Avg. length | - | 4.30 | 4.31 | 3.90 |

are commercially available, and intermediate reactants require being synthesized from the building blocks. In this challenging task, we demonstrate how our method could be used to improve the existing template-free Transformer model (TF, Chen et al. 2019). Given a target product, the hybrid algorithm operates as follows: (1) our RETCL proposes a set of reactants from the candidates $\mathcal{C}$; (2) TF proposes additional reactants outside the candidates $\mathcal{C}$; (3) TF chooses the top-$K$ reactants based on its log-likelihood of all the proposed reactants. As an additional baseline, we replace RETCL by another independently trained TF in the hybrid algorithm. We use Retro* (Chen et al., 2020a) for efficient route search with the retrosynthesis models and evaluate the discovered routes based on the metrics used by Kishimoto et al. (2019); Chen et al. (2020a). As reported in Table 6, our model can enhance the search quality of the existing template-free model in the multi-step retrosynthesis scenarios. This is because our RETCL is enable to recommend available and plausible reactants to TF for each retrosynthesis step. Note that the MLP column is the same as reported in Chen et al. (2020a) which uses a template-based single-step MLP model. The detailed description of this multi-step retrosynthesis experiment and the discovered routes are provided in Appendix D.

## 4 RELATED WORK

The template-based approaches (Coley et al., 2017b; Segler & Waller, 2017; Dai et al., 2019) rely on reaction templates that are extracted from a reaction database (Coley et al., 2017a; 2019) or encoded by experts (Szymkuć et al., 2016). They first select one among known templates, and then apply it to the target product. On the other hand, template-free methods (Liu et al., 2017; Karpov et al., 2019; Zheng et al., 2019; Shi et al., 2020) consider retrosynthesis as a conditional generation problem such as machine translation. Recently, synthon-based approaches (Shi et al., 2020; Somnath et al., 2020) have also shown the promising results based on utilizing the atom-mapping between products and reactants as an additional type of supervisions.

Concurrent to our work, Guo et al. (2020) also propose a selection-based approach, Bayesian-Retro, based on sequential Monte Carlo sampling (Del Moral et al., 2006). As reported in Table 1, our RETCL significantlly outperforms Bayesian-Retro. The gap is more evident since it uses $6 \times 10^5$ forward evaluations (i.e., 6 hours[3]) of Molecular Transformer (Schwaller et al., 2019a) for single-step retrosynthesis of one target product while our RETCL requires only one second.

## 5 CONCLUSION

In this paper, we propose RETCL for solving retrosynthesis. To this end, we reformulate retrosynthesis as a selection problem of commercially available reactants, and propose a contrastive learning scheme with hard negative mining to train our RETCL. Through the extensive experiments, we show that our framework achieves outstanding performance for the USPTO benchmarks. Furthermore, we demonstrate the generalizability of RETCL to unseen reaction templates. We believe that extending

---

[3]See https://github.com/zguo235/bayesian_retro.

our framework to multi-step retrosynthesis or combining with various contrastive learning techniques in other domains could be interesting future research directions.

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

# A   DATASET DETAILS

We here describe the details of USPTO datasets. The reactions in the USPTO datasets are derived from the US patent literature (Lowe, 2012). The entire set, USPTO 1976-2016, contains 1.8 million raw reactions. The commonly-used benchmark of single-step retrosynthesis is USPTO-50k containing 50k clean atom-mapped reactions which can be classified into 10 broad reaction types (Schneider et al., 2016). See Table 7a for the information of the reaction types. For generalization experiments in Section 3.4, we introduce a filtered dataset, USPTO-50k-modified, which contains reactions of reaction types from 1 to 5. We report the number of reactions of the modified dataset in Table 7b. We also use the USPTO-full dataset, provided by Dai et al. (2019), which contains 1.1 million reactions. Note that we use only the test split of USPTO-full (i.e., only 101k reactions) for testing generalizability. Note that we do not use atom-mappings in the USPTO benchmarks. Moreover, we do not consider reagents for single-step retrosynthesis following prior work (Liu et al., 2017; Dai et al., 2019; Lin et al., 2019; Karpov et al., 2019).

Table 7: The detailed information on USPTO datasets.

(a) The information about reaction types in USPTO-50k.

| ID | Fraction (%) | Description |
|----|------|-------------|
| 1 | 30.3 | heteroatom alkylation and arylation |
| 2 | 23.8 | acylation and related processes |
| 3 | 11.3 | C-C bond formation |
| 4 | 1.8 | heterocycle formation |
| 5 | 1.3 | protections |
| 6 | 16.5 | deprotections |
| 7 | 9.2 | reductions |
| 8 | 1.6 | oxidations |
| 9 | 3.7 | functional group interconversion (FGI) |
| 10 | 0.5 | functional group addition (FGA) |

(b) The number of reactions in USPTO datasets.

| Dataset | Split | # of reactions |
|---------|-------|----------------|
| USPTO-50k | Train | 40,008 |
| | Val | 5,001 |
| | Test | 5,007 |
| USPTO-50k-modified | Train | 27,429 |
| | Val | 3,429 |
| | Test | 5,007 |
| USPTO-full | Train | 810,496 |
| | Val | 101,311 |
| | Test | 101,311 |

## B  IMPLEMENTATION DETAILS

We here provide a detailed description of our implementation. Since the USPTO datasets provide molecule information based on the SMILES (Weininger, 1988) format, we convert a SMILES representation to a bidirectional graph with atom and bond features. To this end, we use RDKit[4] and Deep Graph Library (DGL) (Wang et al., 2019). Let $G = (V, E)$ be the molecular graph, and $X(v) \in \mathbb{R}^{d_{\text{atom}}}$ and $X(uv) \in \mathbb{R}^{d_{\text{bond}}}$ are features for an atom $v \in V$ and a bond $uv \in E$ in the molecular graph $G$, respectively. The atom feature $X(v)$ includes the atom type (e.g., C, I, B), degree, formal charge, and so on; the bond feature $X(uv)$ includes the bond type (single, double, triple or aromatic), whether the bond is in a ring, and so on. For more details, we highly recommend to see DGL and its extension, DGL-LifeSci.[5]

**Architecture.** We build our graph neural network (GNN) architecture based on the molecular graph $G$ with features $X$ as follows:

$$H^{(0)}(v) \leftarrow \texttt{ReLU}\left(\texttt{BN}\left(W_{\texttt{atom}}^{(0)} X(v) + \sum\nolimits_{u \in \mathcal{N}(v)} W_{\texttt{bond}}^{(0)} X(uv)\right)\right),$$

$$H^{(l)}(v) \leftarrow \texttt{ReLU}\left(\texttt{BN}\left(W_1^{(l)} \sum\nolimits_{u \in \mathcal{N}(v)} H^{(l-1)}(u) + \sum\nolimits_{u \in \mathcal{N}(v)} W_{\texttt{bond}}^{(l)} X(uv)\right)\right),$$

$$H^{(l)}(v) \leftarrow \texttt{ReLU}\left(\texttt{BN}\left(W_2^{(l)} H^{(l)}(v) + H^{(l-1)}(v)\right)\right), \text{ for } l = 1, 2, \ldots, L,$$

$$H(v) \leftarrow W_{\texttt{last}} H^{(L)}(v),$$

where $\mathcal{N}(v)$ is the set of adjacent vertices with $v$. This architecture is based on structure2vec (Dai et al., 2019; 2016), however it is slightly different with the model used by Dai et al. (2019): we use ReLU after BN instead of BN after ReLU; we append a last linear model $W_{\texttt{last}}$. Based on the atom-level embeddings $H(v)$, we construct query and key embeddings $f_\theta$, $g_\theta$, and $h_\theta$ using three separate residual blocks as follows:

$$f_\theta(M) \leftarrow \sum_{v \in V} \left(H(v) + \texttt{BN}(W_2^{(f)}\texttt{ReLU}(\texttt{BN}(W_1^{(f)}\texttt{ReLU}(H(v)))))\right),$$

$$g_\theta(M) \leftarrow \sum_{v \in V} \left(H(v) + \texttt{BN}(W_2^{(g)}\texttt{ReLU}(\texttt{BN}(W_1^{(g)}\texttt{ReLU}(H(v)))))\right),$$

$$h_\theta(M) \leftarrow \sum_{v \in V} \left(H(v) + \texttt{BN}(W_2^{(h)}\texttt{ReLU}(\texttt{BN}(W_1^{(h)}\texttt{ReLU}(H(v)))))\right),$$

where $M$ is the corresponding molecule with the molecular graph $G$. Note that $\theta$ includes all $W$ defined above, and we omit bias vectors of the linear layers due to the notational simplicity. We found that these design choices, e.g., sharing GNN layers and using residual layers, also provide an accuracy gain. Therefore, more sophisticated architecture designs could provide further improvements; we leave it for future work.

**Optimization.** For learning the parameter $\theta$, we use the stochastic gradient descent (SGD) with a learning rate of $0.01$, a momentum of $0.9$, a weight decay of $10^{-5}$, a batch size of $64$, and a gradient clip of $5.0$. We train our model for $200k$ iterations and evaluate on the validation split every $1000$ iterations. The information of the nearest neighbors is also updated every $1000$ iterations. When evaluating on the test split, we use the best validation model with a beam size of $200$.

To sum up, we use Pytorch (Paszke et al., 2017) for automatic differentiation, Deep Graph Library (Wang et al., 2019) for building graph neural networks, and RDKit for processing SMILES (Weininger, 1988) representations. All our models can be executed on single NVIDIA RTX 2080 Ti GPU.

---

[4]Open-Source Cheminformatics Software, https://www.rdkit.org/
[5]Bringing Graph Neural Networks to Chemistry and Biology, https://lifesci.dgl.ai/

## C  FAILURE CASE STUDY

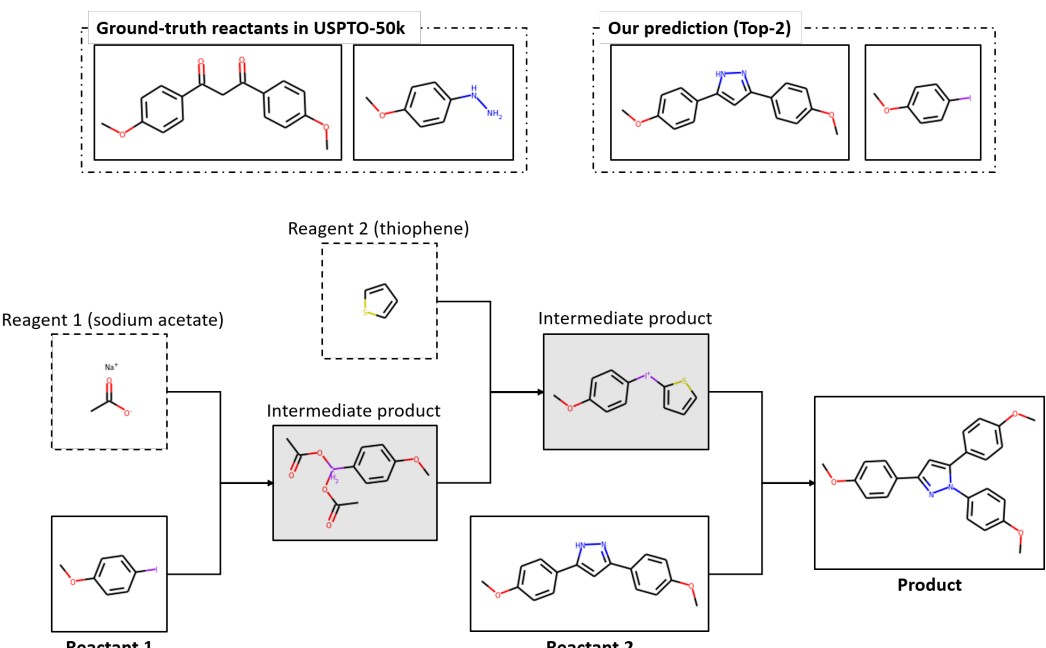

Figure 5: A synthetic path existing in Reaxys based on RETCL's prediction.

As illustrated in Figure 5, we found that our RETCL's prediction differs from the ground-truth reactants in USPTO-50k, however, it exists as a 3-step reaction with two reagents (sodium acetate and thiophene) in the chemical literature (Gonda & Novak, 2015).[6] Note that our framework currently does not consider reagent prediction. Therefore, our prediction can be regarded as an available (i.e., correct) synthetic path in practice.

---

[6]We found this synthetic path and the corresponding literature from a chemical database, Reaxys. Note that the sodium acetate and the thiophene are considered as reagents in Reaxys.

# D    MULTI-STEP RETROSYNTHESIS

For the multi-step retrosynthesis experiment described in Section 3.4, we use a synthetic route dataset provided by Chen et al. (2020a). This dataset is constructed from the USPTO (Lowe, 2012) database like other benchmarks. We recommend to see Chen et al. (2020a) for the construction details. The dataset contains 299202 training routes, 65274 validation routes, and 190 test routes. We first extract single-step reactions and molecules from the training and validation splits of the dataset. The extracted reactions are used for training our RETCL and Transformer (TF, Chen et al. 2019), and the molecules are used as the candidate set $\mathcal{C}_{\texttt{train}}$[7] for ours. When testing the single-step models with Retro* (Chen et al., 2020a), we use all starting molecules (i.e., 114802 molecules) in the routes in the dataset as the candidate set $\mathcal{C}$. This reflects more practical scenarios because intermediate reactants often be unavailable in multi-step retrosynthesis. We remark that TF also uses the candidate set $\mathcal{C}$ as the prior knowledge for finishing the search procedure.

The evaluation metrics used in Section 3.4 are *success rate* and *average length of routes*. The success means that a synthetic route for a target product is successfully discovered under a limit of the number of expansions. We set the limit by 100 and use only the top-5 predictions of a single or hybrid model for each expansion. When computing the average length, we only consider the cases where all the single-step models discover routes successfully. As Chen et al. (2020a) did, we use the negative log-likelihood computed by TF as the reaction cost.

Figure 6 and 7 illustrate the discovered routes by TF and RETCL+TF under the aforementioned setting. The molecules in the blue boxes are building blocks (i.e., available reactants) and the numbers indicate the reaction costs (i.e., the negative log-likelihoods computed by TF). As shown in the figures, our algorithm allows to discover a shorter and cheaper route.

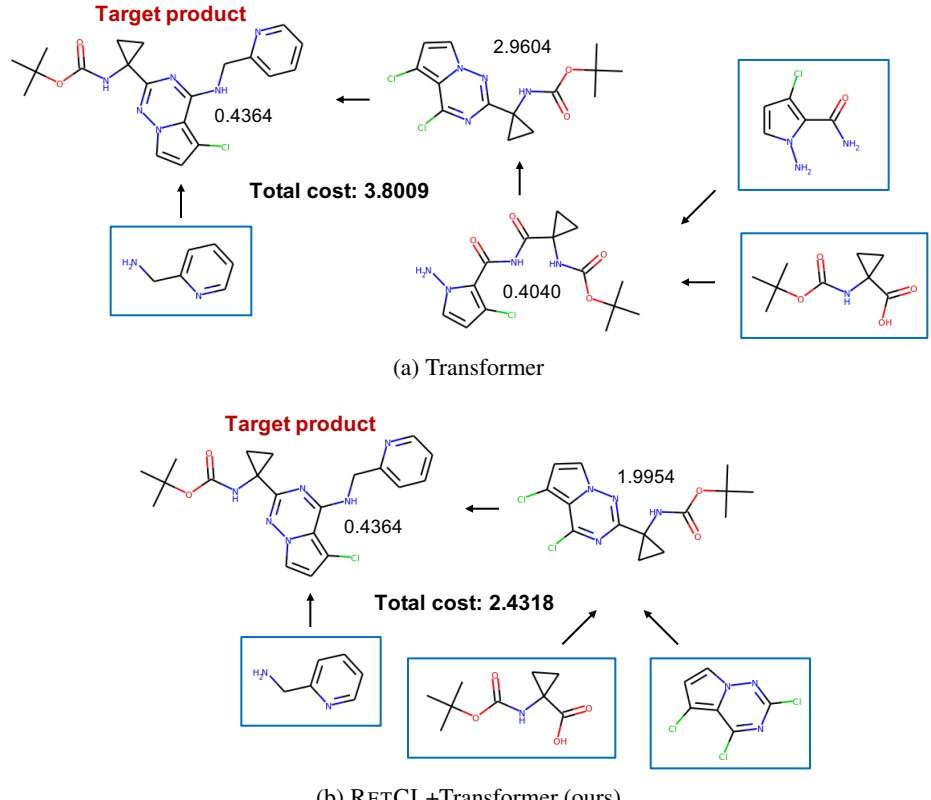

(a) Transformer

(b) RETCL+Transformer (ours)

Figure 6: Synthetic routes discovered by (a) Transformer and (b) our RETCL+Transformer.

---

[7]Note that this candidate set is used only for training.

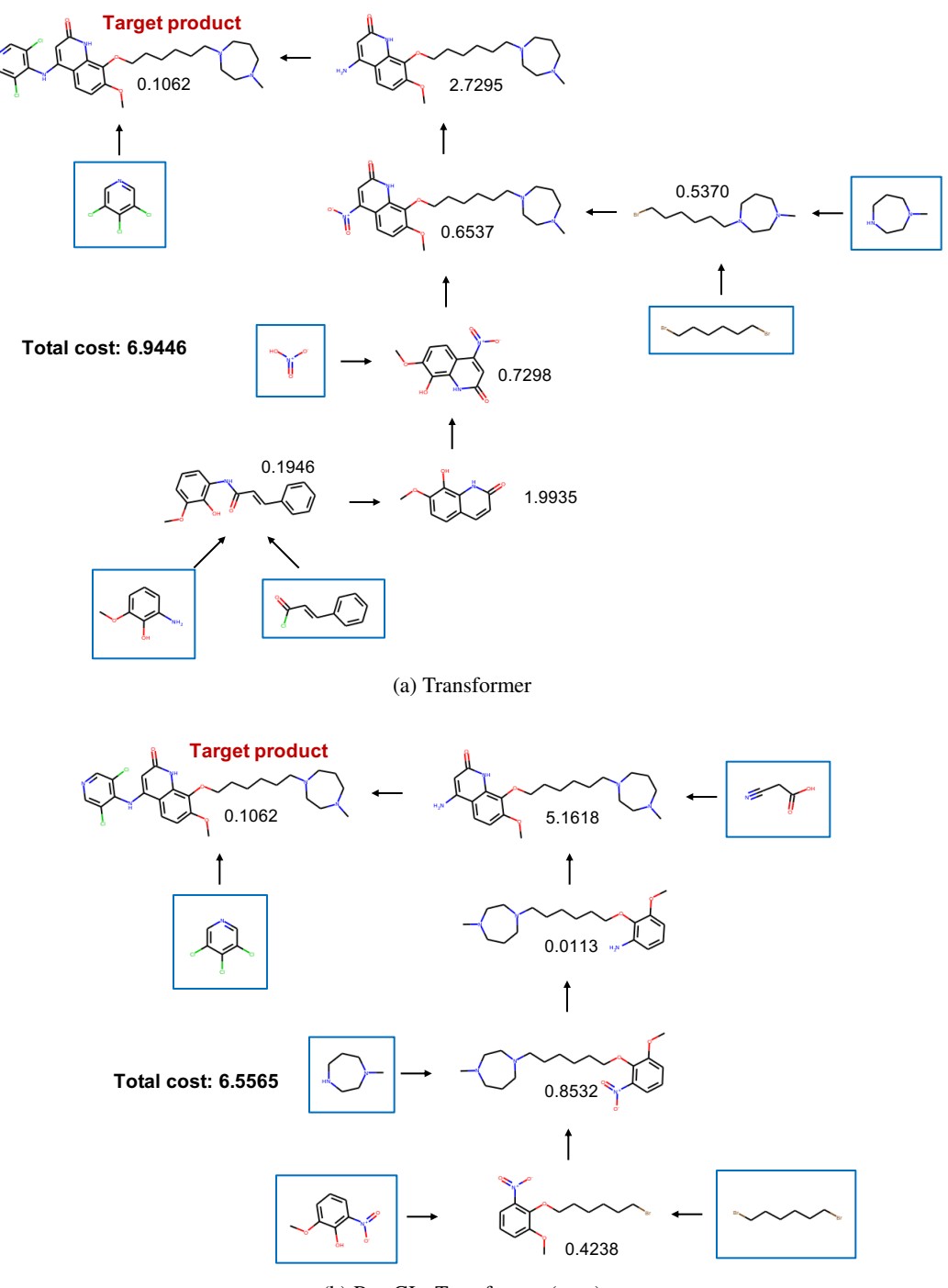

Figure 7: Synthetic routes discovered by (a) Transformer and (b) our RETCL+Transformer.

## E   GENERALIZATION TO UNSEEN CANDIDATES

The knowledge of the candidate set $\mathcal{C}$ could be updated after learning the RETCL framework. In this case, the set used in the test phase is larger than that in the training phase, i.e., $\mathcal{C}_{\mathtt{train}} \subsetneq \mathcal{C}_{\mathtt{test}}$. One can learn the framework once again, however someone want to use it instantly without additional training. To validate that our framework can generalize to unseen candidates, we conduct an additional experiments with a smaller candidate set $\mathcal{C}_{\mathtt{small}}$. We first train our model with $\mathcal{C}_{\mathtt{train}} = \mathcal{C}_{\mathtt{small}}$ and then test with the larger candidate set $\mathcal{C}_{\mathtt{test}} = \mathcal{C}_{\mathtt{large}}$. Here we consider two cases of $\mathcal{C}_{\mathtt{small}}$: (a) 91k molecules in training and validation splits of USPTO-50k; (b) 100k molecules in all splits of USPTO-50k. As reported in Table 8, the model trained with $\mathcal{C}_{\mathtt{small}}$ achieves comparable performance to the model trained with $\mathcal{C}_{\mathtt{large}}$. This demonstrates that our model trained with a small corpora (e.g., USPTO-50k) can work with unseen candidates.

Table 8: Generalization to unseen candidates.

| $|\mathcal{C}_{\mathtt{train}}|$ | $|\mathcal{C}_{\mathtt{test}}|$ | Top-1 | Top-3 | Top-5 | Top-10 | Top-20 | Top-50 |
|---|---|---|---|---|---|---|---|
| 671,518 | | 71.3 | 86.4 | 92.0 | 94.1 | 95.0 | 96.4 |
| 100,508 | 671,518 | 68.8 | 84.1 | 87.6 | 90.0 | 91.8 | 93.5 |
| 91,297 | | 69.0 | 84.9 | 88.1 | 91.0 | 92.8 | 94.4 |

# F    RESTRICTION OF KNOWLEDGE OF CANDIDATE REACTANTS

One might have very restricted knowledge of the candidate set $\mathcal{C}$ of commercially available reactants due to own circumstances such as a budget limit. In this case, one of ground-truth reactants might be missing from the candidate set. Since there exist multiple solutions in the retrosynthesis task, retrosynthesis tools should be able to provide alternative solutions in such a case. To verify that our RETCL can recommend such a solution, we experiment with varying sizes of the candidate set $\mathcal{C}$. Remark that using a smaller candidate set means only a small number of reactants is practically available, thus performance degradation is expected. Here, we use two metrics: exact-match accuracy and coverage proposed by Schwaller et al. (2019b). To be specific, the coverage measures whether the target product is synthesized from the predicted reactant-set made by RetCL using a forward synthesis model (Schwaller et al., 2019a). This can evaluate plausibility of the prediction even if one of the reactants is missing from the candidate set. We report the metrics using top-10 predictions in the USPTO-50k test split. As shown in Table 9, even though some ground-truth candidates are missing, our framework can provide plausible solutions.

Table 9: Top-10 exact-match accuracy (%) and coverage (%) under restricted knowledge of the candidate set $\mathcal{C}$ of commercially available reactants.

| Subset size | 25% | 50% | 75% | 100% |
|---|---|---|---|---|
| Exact-match | 10.2 | 29.9 | 57.9 | 93.8 |
| Coverage | 25.3 | 52.1 | 72.7 | 89.0 |
| Exact-match or Coverage | 27.4 | 56.8 | 79.3 | 97.9 |

