# OpenReview forum: "RetCL: A Selection-based Approach for Retrosynthesis via Contrastive Learning"
_ICLR.cc/2021/Conference — Reject_

### Official Review · AnonReviewer1 · 2020-10-25
**Clever contrastive learning, but the formulation and experiments are problematic**

**Rating:** 4
**Confidence:** 5

**Review:**

This submission describes an approach to single-step retrosynthesis based on contrastive learning that selects reactants that can be used to synthesize a target product in a single step. The stated contributions are (1) an approach to retrosynthesis that is constrained to only select “available” starting materials; and (2) a novel contrastive learning scheme with hard negative mining. The use of a contrastive loss to learn an embedding of reactants and their products that exhibit the property that the sum of reactant vectors has a high cosine similarity to the product vector is clever. This is a nice way to give structure to a continuous vector space. The strategy of hard negative mining is also clever and identifies the examples that would intuitively be the most informative.

I take some issue with the premise of constraining retrosynthetic recommendations to an enumerated list. This works for small corpora like the ones used here, but in reality, retrosynthesis is a multi-step process where most reactants are *not* commercially available and the one-step retrosynthetic expansion must be repeated recursively. This approach is fundamentally unable to operate on reaction products where multiple synthetic steps are required, which represent the challenging cases. That is why in the multi-step evaluations, the authors rely on the Transformer model to propose intermediate structures. Appendix D also suggests that in the pathway search experiments, knowledge of the routes in advance was required to construct the set of all starting materials to select from.

The empirical evaluation, as a result of the premise, is somewhat flawed. By constraining reactant proposals to an enumerated list of reactants extracted from a parent database, the authors have simplified the task in comparison to previous approaches, making a head-to-head comparison of accuracy less informative. The evaluation in 3.4 generalizing to unseen templates could simply be an indication that the model is learning atom conservation (excepting leaving groups) and to maximize substructure overlap with the products. The model has access to a restricted list of possible starting materials of which very few are likely to be plausible precursors for a given product. This advantage invalidates the comparison in Table 5; this is not evidence of generalization, since the test set answers were included in the set of candidates.

While the contrastive learning approach is clever, this work uses a contrived formulation for retrosynthesis that is not applicable to multi-step planning and the experiments do not support the conclusions drawn.

---

> ### Author Response · Authors · 2020-11-18
> **Response to Reviewer #1**
>
> We sincerely appreciate your efforts and constructive comments. We address your comments one by one.
>
> **Comment:** Concern about the premise of constraining recommendations to an enumerated list. \
> **Response:** Thank you for mentioning an important point. Our premise is crucial for retrosynthesis models to be practical because the starting materials need to be commercially available to execute a synthetic route of multiple reactions. Although our RetCL cannot solve the multi-step retrosynthesis task alone, it still provides a meaningful contribution: RetCL can be combined with an existing retrosynthesis model to yield the best performance with respect to the given starting materials (as shown in Section 3.4 and Appendix D).
>
> ---
>
> **Comment:** Knowledge of synthetic routes is required to construct the candidate set of starting materials. \
> **Response:** The knowledge is not required to construct the candidate set. Our choice is necessary to construct synthesizable products only for evaluating the performance of RetCL. Note that a similar choice has been made by Bradshaw et al. [1] and Chen et al. [2]. In general, our RetCL can be applied to any set of starting materials without the knowledge of synthetic routes.
>
> [1] Bradshaw et al., A Model to Search for Synthesizable Molecules, NeurIPS 2019 \
> [2] Chen et al., Retro*: Learning Retrosynthetic Planning with Neural Guided A* Search, ICML 2020
>
> ---
>
> **Comment:** A head-to-head comparison is less informative since the task is over-simplified. \
> **Response:** As mentioned in Section 3.2, we agree that the additional knowledge of starting materials simplifies the retrosynthetic task and for that reason we do not make a head-to-head comparison. Furthermore, from Table 2, one can verify how the additional knowledge does not over-simplify the task. To be specific, Table 2 shows how one can incorporate the additional knowledge into the baselines. Here one can see that the knowledge is not significant enough to conclude that the task has been over-simplified, e.g., Transformer achieves 59.6% accuracy while RetCL achieves 71.3%. This also validates how our RetCL learns chemical rules more sophisticated than simple atom preservation and maximization of substructure overlap.

---

### Official Review · AnonReviewer2 · 2020-10-27

**Rating:** 4
**Confidence:** 5

**Review:**

### Summary of the paper
This paper proposes a sequential reactant selection scheme for retrosynthesis. In each step, the model gives a ranking of reactants based on previously chosen reactants $R_{given}$. After all the reactants are selected, the model checks whether the chosen reactants result in desired product. The ranking module $\psi$ is trained via contrastive learning. The negative reactant candidates are constrained to be similar to the positive reactant.

### Strength and weakness
1. The method proposes a selection based approach to address one of the weaknesses of template-free approaches -- the predicted reactants may be commercially unavailable. This is indeed an important issue that needs to be addressed. However, I am afraid the proposed approach is an overkill. If we only select reactants appeared in the USPTO database, the model cannot generalize to new reactions which involves new reactants not in the USPTO database. This is problematic for two reasons:
 1. For instance, if the model is evaluated on a harder test set where ground truth reactants are not in USPTO database, I think the model will fail (with 0% top-1 accuracy).
 2. Moreover, in multi-step retrosynthesis, you are allowed to make new intermediate compounds from commercially available compounds in order to make your final product. I don't see why we have to choose reactants only from commercially available compounds.
2. Scalability: The total number of commercially available compounds are up to $10^9$ (e.g., Enamine REAL database). I am concerned that the neural network based ranking will run very slowly and cannot scale to larger sets of compounds.
3. In section 3.4, authors evaluate their approach on a harder test set with novel reaction templates. In my opinion, template-free approaches are also capable of generalizing to novel reaction templates. Why there is no comparison to G2G or transformer in table 5?
4. The proposed method is quite straightforward, with limited technical novelty in my opinion. I am afraid the contrastive learning part is just a straightforward application of negative sampling.
5. The result looks very strong on the USPTO-50K test set.

### Overall evaluation
I vote for weak reject mainly for two reasons.
1. The method seems incapable of generalizing to new reactant compounds outside of the commercial library.
2. The approach has limited novelty to ICLR audience.

### Post rebuttal
I would like to thank the authors for their valuable response. The experimental results seem strong, but technical novelty is still limited. My review score remains the same. I believe this paper can make great impact if submitted to a chemistry journal.

### Suggestions
Despite my negative evaluation, I do appreciate the point authors are trying to address. I think the paper can be significantly strengthened if you can loosen the "commercially available" constraint. For instance, you can loosen the constraint to "synthesizable" -- choosing the reactants that can be synthesized from a given set of building blocks (via reactions)! In multi-step retrosynthesis, you are allowed to make new compounds from your building blocks for the sake of making your final products.

---

> ### Author Response · Authors · 2020-11-18
> **Response to Reviewer #2**
>
> We sincerely appreciate your efforts and constructive comments. We address your comments one by one.
>
> **Comment:** The method seems incapable of generalizing to new reactant compounds outside of the commercial library. This paper can be significantly strengthened if loosen the "commercially available" constraint, e.g., choosing the reactants that can be synthesized from a given set of building blocks. \
> **Response:** In Section 3.4 and Appendix D, we show how to loosen the “commercially available” constraint by combining our RetCL with an existing Transformer model. In this experiment, the candidate molecules are commercially available building blocks, as described in Appendix D, thus intermediate products may not be in the candidate set. As shown in Table 6, Figure 6, and Figure 7, our RetCL can improve multi-step retrosynthesis based on complementing the Transformer for “commercially available” building blocks.
>
> ---
>
> **Comment:** Scalability with respect to the number of commercially available compounds. \
> **Response:** As you mentioned, the number of available molecules could be very large. However, our search procedure is very effective because it requires only a few matrix-vector multiplications: once the embeddings $g_\theta(R)$ and $h_\theta(R)$ for all candidates $R\in\mathcal{C}$, we only require $f_\theta(P)$ and $h_\theta(P)$ to compute the score functions $\psi$ and $\phi$. Moreover, computational complexity of the procedure can be improved by hierarchical classification techniques.
>
> ---
>
> **Comment:** Why is there no comparison to template-free approaches in table 5? \
> **Response:** As you suggested, we add a template-free baseline (Transformer) for the experiments. As reported below, our framework still outperforms the template-free approaches. This is also reported in Table 5 of our revised paper.
>
> | | top-1 | top-10 | top-50 |
> |-|-|-|-|
> | Transformer | 29.9 | 46.6 | 51.0 |
> | GLN | 26.7 | 42.2 | 46.7 |
> | RetCL (ours) | 39.9 | 57.1 | 60.9|
>
> ---
>
> **Comment:** The proposed method is quite straightforward, with limited technical novelty. \
> **Response:** We think that we proposed various non-trivial components for selection-based retrosynthesis. First, using both forward and backward directions can be considered as a cycle consistency constraint which is typically utilized in domain translation tasks. We show it is also effective in the retrosynthesis problem (see Table 3). Second, the contrastive learning scheme for product-reactant pairs is novel in a way that recent contrastive/self-supervised learning methods [1,2] for molecules do not focus on the relationship between two molecules. Furthermore, we demonstrate that hard-negative sampling is simple but very effective (roughly 10% gain in top-1 accuracy), as shown in Table 3. To our knowledge, considering hard-negatives is explored not enough in contrastive learning for graphs/molecules. Therefore, we believe that our framework is technically novel in various ways, and it is also closely related to other interesting ML topics such as graph/molecule contrastive learning.
>
> [1] Rong et al., Self-Supervised Graph Transformer on Large-Scale Molecular Data, NeurIPS 2020 \
> [2] You et al., Graph Contrastive Learning with Augmentations, NeurIPS 2020

---

### Official Review · AnonReviewer4 · 2020-10-28

**Rating:** 4
**Confidence:** 5

**Review:**

RETCL enumerates all of the candidate molecules (all US patent dataset, all 671k) based on selection scores computed by graph neural networks. The cosine similarity between products and reactants are used to design scores, which is later used for training. The way of ‘cosine similarity to bridge reactants to products are interesting.



Q1:  Used test data as selection set during training

Training using all US patent dataset (671k) as candidates for RETCL to select from leaked test data (USPTO50K test data) during training. All US patent dataset is a superset of the USPTO50K data.  “Selection based algorithm” tends to achieve overly optimistic results due to this reason.

Q2: How to generalize?
It is great the paper shows that RETCL generalizes well to unseen templates
However, if we select upon an existing dataset (though very large). How does RETCL possibly yield totally unseen reactants (i.e. not in any existing dataset)?


Q3 Approximation of C
Sec 2.3
Computing p(R|P, Rgiven, C)  and q(P|R, C)  requires summing over all candidates in C, which is necessary for proper probability, but computational expensive.
How does these probabilities be approximated by a mini-batch of reactions?
The batch statistics are very different from the C (all US patent dataset, all 671k)

---

> ### Author Response · Authors · 2020-11-18
> **Response to Reviewer #4**
>
> We sincerely appreciate your efforts and constructive comments. We address your comments one by one.
>
> **Comment:** Used test data as selection set during training. \
> **Response:** Using the candidate set during training does not lead to overly optimistic results. To verify this, we allow our model to use only a small number of candidates (i.e., molecules in training/validation splits of USPTO-50k) during the training phase. Note that the number of candidates is 91k, and that of all molecules in USPTO-50k is 100k. When testing, we use all 671k molecules. As reported below, training without test candidates achieves comparable performance to the case when using all 671k molecules during training. This demonstrates that our results are not overly optimistic. We add this discussion in Appendix E of revision.
>
> | $\mathcal{C}_\mathtt{train}$ | $\mathcal{C}_\mathtt{test}$ | top-1 | top-3 | top-5 | top-10 | top-20 | top-50|
> |-|-|-|-|-|-|-|-|
> | 671k | 671k | 71.3 | 86.4 | 92.0 | 94.1 | 95.0 | 96.4 |
> | 100k | 671k | 68.8 | 84.1 | 87.6 | 90.0 | 91.8 | 93.5 |
> | 91k | 671k | 69.0 | 84.9 | 88.1 | 91.0 | 92.8 | 94.4 |
>
> ---
>
> **Comment:** How does RetCL possibly yield totally unseen reactants? \
> **Response:** Thank you for mentioning an important point. As we show in Section 3.4 and Appendix D, our RetCL can be combined with a retrosynthesis model to yield unseen reactants. The totally unseen reactants fall into two cases depending on synthesizability from a set of commercially available building blocks. If the reactant is synthesizable, our hybrid model can find such a route (see Section 3.4 and Appendix D). Otherwise, discovery is practically meaningless because the final target product cannot be synthesized in practice even though a route is discovered. We remark that our goal is to discover a synthetic route starting from commercially available reactants for synthesizing a target product.
>
> ---
>
> **Comment:** Approximation of summation over all candidates in $\mathcal{C}$. \
> **Response:** Since batch statistics are very different from $\mathcal{C}$ as you mentioned, we add hard-negative samples into $\mathcal{C}_\mathcal{B}$ for better approximation (see $\tilde{\mathcal{C}}_\mathcal{B}$). The hard-negative samples have high cosine similarities by definition, thus they often amount to the majority of the summation. Hence, using $\tilde{\mathcal{C}}_\mathcal{B}$ improves the approximation. As reported in Table 3, we show that better approximation improves selection qualities.

---

### Official Review · AnonReviewer3 · 2020-10-31
**How does this extend to the real world task?**

**Rating:** 5
**Confidence:** 4

**Review:**

This paper poses an approach to retrosynthesis that addresses the challenges of (i) availability of reactants and (ii) generalization to unseen templates. To achieve this the authors reformulate retrosynthesis as the selection of reactants from a fixed set, in the case of the USPTO database this is the set of 671,578 commercially available reactants used in the database. Their reactant selection framework RetCL uses GNNs to calculate selection scores for all candidate molecules. A novel contrastive training scheme is used to learn the selection score, which is computed as the cosine similarity between embeddings of the product and the reactants computed by the graph neural networks. The authors provide a good summary of related work in this area.

My major questions for the authors of this paper concern the limitation imposed by restricting to a specific candidate set. For example, the results reported on USPTO-50k are very impressive, particularly the ability to generalize to heldout reaction types. However, the model is never challenged by reactions for which the reactants are not present in the candidate set, and it is completely unclear how the model would perform in this scenario. This is important, because to tackle the chemically relevant retrosynthesis problem it is exactly necessary to solve reactions where the reactants may not be present in the 671,518 reactants present in USPTO-50k.

I find it a bit surprising that the authors do not address this constraint in the main text, and I would ask that they carry out experiments where they restrict the candidate set to a subset of the 671,518 reactants yet consider reactions from the whole database, and ask how severe this restriction is in terms of the solutions obtained by the model (it would be necessary to build these splits carefully, and provide results for multiple splits). There are other arguments that the authors could also make to defend this point in addition to these experiments - but the current state in which the issue is ignored is not satisfactory.

The description of the contrastive training approach is clear and coherent. The addition of hard negatives to the batches to improve learning is interesting, and the results of the ablation study speak to the important role that it plays.

---

> ### Author Response · Authors · 2020-11-18
> **Response to Reviewer #3**
>
> We sincerely appreciate your efforts and constructive comments. We address your comments one by one.
>
> **Comment:** Concern about the limitation imposed by restricting to a specific candidate set. \
> **Response:** Thank you for mentioning an important point. We tackle the limitation (of restricting to a specific candidate set) in Section 3.4 and Appendix D. As you said, a reactant may not be present in the candidate set. This can be considered as two cases depending on synthesizability from a set of commercially available building blocks (i.e., the candidate set). To tackle the case where the reactant is synthesizable, our hybrid model can be applied as shown in Section 3.4 and Appendix D. Note that we do not consider the other case because it cannot be synthesized in the real world, i.e., practically meaningless (due to the unavailability of building blocks).
>
> ---
>
> **Comment:** Additional experiments restricting the candidate set. \
> **Response:** To alleviate your concern, we experiment with varying sizes of the candidate set. We remark that using a smaller candidate set means that only a small number of reactants is practically available, thus performance degradation is expected. In this experiment, we use two metrics: exact-match accuracy and coverage [1]. To be specific, the coverage measures whether the target product is synthesized from the predicted reactant-set made by RetCL (using a forward synthesis model, Molecular Transformer [2]). This can evaluate plausibility of the prediction even if one of the reactants is missing from the candidate set. We report the metrics using top-10 predictions in the USPTO-50k test split. As shown in the table below, even though we use a small number of candidates, our framework can provide plausible solutions. We add this discussion in Appendix F of the revised paper.
>
> | Subset size | 25% | 50% | 75% | 100% |
> |-|-|-|-|-|
> | Exact-match | 10.2 | 29.9 | 57.9 | 93.8 |
> | Coverage | 25.3 | 52.1 | 72.7 | 89.0 |
> | Exact-match or Coverage | 27.4 | 56.8 | 79.3 | 97.9 |
>
> [1] Schwaller et al., Evaluation Metrics for Single-Step Retrosynthetic Models, NeurIPS 2019 Workshop on Machine Learning and the Physical Science \
> [2] Schwaller et al., Molecular Transformer: A Model for Uncertainty-Calibrated Chemical Reaction Prediction, ACS Cent. Sci. 2019

---

### Author Response · Authors · 2020-11-18
**Common Response**

Dear all reviewers,

First of all, we thank all reviewers for your efforts  in providing valuable comments on our manuscript.

**[Clarification of our motivation]** Many reviewers expressed concerns about our model restricting the candidate set of reactants. In response, we clarify that our restriction is a practical requirement rather than an additional knowledge used to achieve high performance. Namely, our work is motivated by how “practically usable” synthetic routes start from commercially available molecules. Therefore, it is natural to assume having access to the candidate set of reactants that are commercially available. We believe such a research direction to play an important role in future research for practical retrosynthetic tasks.

---

**[Revision]** We have revised and enhanced our manuscript with the following additional experiments and discussions:
- an ablation study of the training candidate set, suggested by R4 (Appendix E, Table 8)
- another baseline in Table 5, suggested by R2 (Table 5)
- an ablation study of smaller candidate sets, suggested by R3 (Appendix F, Table 9)

The revisions made are marked with “red” in the revised manuscript.

Thank you very much, \
Authors

---

### Decision · Program_Chairs · 2021-01-07
**Final Decision**

**Decision:**

Reject

**Comment:**

While the authors appreciated the proposed contrastive training scheme and the strong related work summary, all authors agreed that the approach was severeley limited by being a pure selection-based method. Without the help of another model that proposes molecules, the approach can only select reactants from an existing set. As target molecules become more complicated, the modeller must make a choice: (a) use a much larger initial candidate set which hopefully encompases all molecules necessary to make the target molecule, or (b) use another model to propose new intermediate molecules. The authors went with (b) which harmed their novelty claim: a big reason why retrosynthesis is hard is because of the need to generate unseen molecules, and if this is left to an already proposed model, the current approach is not adding much methodological novelty. While their approach does improve upon existing work in the multi-step setting, there's even more recent work that has not been compared against (e.g., https://arxiv.org/pdf/2006.07038.pdf) so the improved performance may be outperformed.

The fix is straightforward: modify the methodology to also propose intermediate molecules. This will fix the novelty complaint and strengthen the practicality argument: practitioners could directly use this approach to discover synthesis routes. The authors could slightly update the related work, add comparisons against recent methods, and take into account the other feedback given by the authors. The paper is very nicely written, the proposed changes are purely methodological, and not insurmountable in my opinion. I would urge the authors to make these changes which I believe will result in a very nice paper.